# Increasing Wear Resistance of UHMWPE by Loading Enforcing Carbon Fibers: Effect of Irreversible and Elastic Deformation, Friction Heating, and Filler Size

**DOI:** 10.3390/ma13020338

**Published:** 2020-01-11

**Authors:** Sergey V. Panin, Lyudmila A. Kornienko, Vladislav O. Alexenko, Dmitry G. Buslovich, Svetlana A. Bochkareva, Boris A. Lyukshin

**Affiliations:** 1Lab. of Mechanics of Polymer Composite Materials, Institute of Strength Physics and Materials Science SB RAS, Tomsk 634055, Russia; rosmc@ispms.ru (L.A.K.); vl.aleksenko@mail.ru (V.O.A.); buslovichdg@gmail.com (D.G.B.); svetlanab7@yandex.ru (S.A.B.); lba2008@yandex.ru (B.A.L.); 2Department of Materials Science, Engineering School of Advanced Manufacturing Technologies, National Research Tomsk Polytechnic University, Tomsk 634030, Russia; 3Department of Mechanics and Graphics, Tomsk State University of Control Systems and Radioelectronics, Tomsk 634050, Russia

**Keywords:** polymer matrix composites, ultrahigh molecular weight PE, carbon fibers, wear, dry sliding friction, computer simulation, elastic recovery, tribological loading, friction heating

## Abstract

The aim of the study was to develop a design methodology for the UltraHigh Molecular Weight Polyethylene (UHMWPE)-based composites used in friction units. To achieve this, stress–strain analysis was done using computer simulation of the triboloading processes. In addition, the effects of carbon fiber size used as reinforcing fillers on formation of the subsurface layer structures at the tribological contacts as well as composite wear resistance were evaluated. A structural analysis of the friction surfaces and the subsurface layers of UHMWPE as well as the UHMWPE-based composites loaded with the carbon fibers of various (nano-, micro-, millimeter) sizes in a wide range of tribological loading conditions was performed. It was shown that, under the “moderate” tribological loading conditions (60 N, 0.3 m/s), the carbon nanofibers (with a loading degree up to 0.5 wt.%) were the most efficient filler. The latter acted as a solid lubricant. As a result, wear resistance increased by 2.7 times. Under the “heavy” test conditions (140 N, 0.5 m/s), the chopped carbon fibers with a length of 2 mm and the optimal loading degree of 10 wt.% were more efficient. The mechanism is underlined by perceiving the action of compressive and shear loads from the counterpart and protecting the tribological contact surface from intense wear. In doing so, wear resistance had doubled, and other mechanical properties had also improved. It was found that simultaneous loading of UHMWPE with Carbon Nano Fibers (CNF) as a solid lubricant and Long Carbon Fibers (LCF) as reinforcing carbon fibers, provided the prescribed mechanical and tribological properties in the entire investigated range of the “load–sliding speed” conditions of tribological loading.

## 1. Introduction

Antifriction polymer composite materials are widely used as parts of friction units and sealing elements of equipment. The former are of crucial importance ensuring their reliability and durability. One of the widespread heavy duty materials is ultra-high molecular weight polyethylene (UHMWPE) since it possesses low friction coefficient as well as high wear- and chemical resistance. For this reason, parts of critical structures [1,2] and up-to-date medical devices [3,4,5,6,7,8,9] have been manufactured from UHMWPE. At the same time, low elastic modulus and melting point, as well as deformability, do limit its use in heavy-duty friction units, especially in the dry friction conditions [10,11,12]. However, UHMWPE strength and functional characteristics can be improved by loading with various fillers. This makes it possible to expand the applications of UHMWPE in medicine, mechanical engineering, mining, oil and gas, and chemical industries, as well as agriculture and other areas of technology [13,14,15,16,17]. For example, UHMWPE-based composites loaded with fibrous fillers as reinforcing components possess improved mechanical and tribological properties [18,19,20].

To date, a lot of research has been carried out on the UHMWPE-based composites. They are aimed at searching for commercially available fillers that provide both high mechanical and tribological properties when operating in the extreme conditions (high loads and sliding speeds, low temperatures, aggressive environments, etc.) [21,22,23]. However, in spite of the large number of the already obtained results, a scientifically based approach to select fillers that improve the specified performance characteristics of the UHMWPE-based composites has not been developed so far. In particular, a correlation between filler types (composition, aspect ratio, effect on permolecular structure formation) and the properties of the UHMWPE-based composites remains not fully clarified. In addition, the dilemma of reversibility and irreversibility of the deformation processes in the subsurface layers under wear tracks of the composites having maximum wear resistance in a wide range of loads and speeds during tribological tests is still not well understood [12].

Numerous experimental studies of the polymers’ frictional behaviors have shown that, under tribological loading, plastic flow in the surface- and subsurface layers occurs. First of all, it is caused by shear stresses transferred from a rotating counterpart. In order to develop a control methodology, it is of importance to evaluate the stress–strain state parameters of the material affecting the dynamics of these processes. For example, the UHMWPE-based composites are loaded with reinforcing fillers that inhibit the deformation development. In addition, in regard to this, computer simulations of the wear processes (including parametric studies) are implemented.

The aim of this work was to develop a design methodology for the UHMWPE-based composites used in friction units. For doing so, stress–strain analysis was conducted using computer simulation of the triboloading processes. In addition, an effect of the carbon fibers size used as reinforcing fillers on formation of subsurface layer structures at tribological contacts as well as composite wear resistance was evaluated.

## 2. Computer Simulation Methodology

In order to assess the effect of carbon fibers of various lengths on the material stress–strain state parameters under dry friction tribological loading, a two-dimensional contact problem of the theory of elasticity [24] was solved. For doing so, the finite element method [25] and the sequential loading procedure [26] were employed. A “block-on-ring” computational scheme of the process were taken corresponding to the experimental research on wearing neat UHMWPE and the UHMWPE-based composites (Figure 1a) [27]. Normal and tangential loads arising during friction were taken into account. Therefore, the physical and geometric nonlinearity of the deformation process was considered. The simulation was performed with the use of the software developed by the authors. Triangular finite elements with six possible movements of nodes were used. The latter ensured the possibility of their application to discretize a region of any shape.

The principle of the virtual work was employed for the solution of the problem [28]. The main idea was to minimize the total potential energy of the system for possible movements of the nodes. As a result, the system of linear equations in the form of an elemental matrix equation for all elements was obtained [26]:(1)$$\left(\left[K^{G}\right]+\left[K\right]\right){\left\{\Delta R\right\}}_{i}={\left\{\Delta R\right\}}_{i}+{\left\{E\right\}}_{i},$$
where i was the number of the loading step at which the movements of the nodes of all elements were calculated; [K^G^] was the stiffness matrix for the state of the initial stresses; [K] was the elementary stiffness matrix [25]; {ΔF}_i_ was the vector of load increments at the next step; {ΔR}_i_ was the vector of movement increments at the nodes; and {E}_i_ was the residual error of the balance of forces in each element [25].

The node displacement values in the grid {U} were determined by the solution of the system of the equations taking into account the boundary conditions. The obtained increments of displacements were summed up with the previous ones. Then, the matrices [K_G_] and [K], as well as the error in the balance of forces, were calculated.

The scheme of a computational domain for modeling the frictional interaction represented a section normal to the friction surface (Figure 1b). The ‘ABCD’ calculation region (Figure 1b) was 5 mm long, 7 mm wide, and 2 mm thick. Displacement along the *x* and *y* axes on the ‘AD’ lower border of the sample was set to zero (Figure 1). Tangent (*t*) and normal stresses (*σ_n_*) on the ‘AB’ and ‘CD’ lateral sample boundaries; on the ‘EGF’ counterpart and ‘BC’ upper sample boundaries (excluding a contact area), as well as on the ‘AB’ and ‘CD’ sides, were equal to zero (Figure 1b). Normal distributed load (*P_n_*) from the ‘EF’ counterpart side acted on the ‘BC’ upper sample boundary. The following condition after each step to ensure the prescribed load values was checked:(2)$$\sum_{i}\left(\sigma_{n\left(i\right)}\cdot \cos\alpha \cdot L_{x}\cdot t\right)\geq P_{n}\pm 5\%,$$
where *i* was the node number of the finite element; σ_n_(i) was stresses in the node i; *L_x_* was length of the finite element border adjacent to the node; α was the inclination angle of the *L_x_* border to the *x*-axis; *t* was the size of the computational domain along the *z*-axis. This equation implied that the sum of the projections of the obtained normal stresses σ_n_ on the *y*-axis in all contacting nodes (on the ‘BC’ sample boundary) must not exceed the specified load with an error of ±5%.

If the load exceeded a specified level, then its value on the ‘EF’ side was reduced and the step was repeated. In addition, the surface-to-surface contact conditions for the ‘BC’ and ‘EGF’ boundaries (Figure 1b) at each step of the load value change were checked. The conditions to avoid intersecting body surfaces were applied in coupled nodes. In general, a contact of each node of one body, not only with a node of another body, but with some point located on an element boundary was permitted [29]. (Figure 2).

If the ‘m’ and ‘l’ nodes of the counterpart (Figure 2, region 2) were taken as the main ones, and the ‘k’ node (Figure 2, region 1) was considered dependent, then its movement via the movement of the main nodes was calculated [29]:(3)$$v_{k}=v_{l}\cdot \left(1-h\right)+h\cdot v_{m},$$
where *V* was movement of the node along the *y*-axis; *h* determined the ‘*k*’ point position on the ‘*lm*’ contact border (0 ≤ *h* ≤ 1):(4)$$h=\sqrt{\frac{{\left(y_{k}-y_{l}\right)}^{2}+{\left(x_{k}-x_{l}\right)}^{2}}{{\left(y_{m}-y_{l}\right)}^{2}+{\left(x_{m}-x_{l}\right)}^{2}}},$$

Then, the equations used for the finite element method were transformed as follows:(5)$$\frac{\partial \Pi }{\partial v_{l}}=\frac{\partial \Pi }{\partial v_{l}}+\frac{\partial \Pi }{\partial v_{k}}\cdot \frac{\partial v_{k}}{\partial v_{l}}=\frac{\partial \Pi }{\partial v_{l}}+\frac{\partial \Pi }{\partial v_{k}}\cdot \left(1-h\right),\;\frac{\partial \Pi }{\partial v_{m}}=\frac{\partial \Pi }{\partial v_{m}}+\frac{\partial \Pi }{\partial v_{k}}\cdot \frac{\partial v_{k}}{\partial v_{m}}=\frac{\partial \Pi }{\partial v_{m}}+\frac{\partial \Pi }{\partial v_{k}}\cdot h,$$
where Π was the functional for the total potential energy:(6)$$\Pi =\int_{v}\left(\sigma_{ij}^{0}\Delta u_{k,i}\delta \Delta u_{k,j}+C_{ijkl}\Delta u_{k,l}\delta \Delta u_{i,j}\right)dV+\int_{v}\Delta T_{i}\delta \Delta u_{i}dS+\;(\int_{v}\sigma_{i,j}^{o}\delta \Delta u_{i,j}dV-\int_{s}T_{i}^{o}\delta \Delta u_{i}dS),$$
where *σ_ij_^0^* is the initial stress tensor components obtained at the previous step; δ(*Δu_i_*) is virtual increments of movements (variations); *δ*(*Δu_k_,_i_*), *δ*(*Δu_k_,_j_*) are derivatives of the movement variations that are components of the linear and nonlinear parts of the strain tensor (δ(Δ*u_i_,_j_*) and *δ*(*Δu_l_,_j_*), respectively); *C_ijkl_* is the elastic modulus tensor coefficients; Δ*T_i_* is the load increment in the nodes on the surface; *T_i_^0^* is the initial load; *S* is the computational domain area; and *V* is the volume of the computational domain.

The above conditions presumed that the system of the Equations (1) was overdetermined. This fact was considered to design the global stiffness matrix of the element as follows. The equation corresponding to the ‘*k*’ dependent node was excluded. Since the stiffness matrix was symmetric, its row and column corresponded to the dependent node movement along the *y*-axis were multiplied by h. Then, they were added to the row and column of the first main node, multiplied by (1–h), added to the row and column of the second main node. The equation elimination was finished by replacing the corresponded row and column in the stiffness matrix with zeros. This caused the system degeneration, which was eliminated by multiplying the diagonal element of the zero line by a “large” number (for example, 10^6^). At that, the free element was set to zero [25]. After combining the matrixes, this procedure gave a deliberately incorrect result that caused zero movement at the node. Therefore, movement at the ‘k’ node after solving the system of the equations was determined by Formula (3).

If the total normal load reached the specified level in the contacted nodes, then the tangential load to the counterpart surface was set according to the Amonton–Coulomb law:(7)$$F_{f}=f\cdot F_{N},$$
where *F_f_* was the friction force, *f* was sliding friction coefficient, and *F_N_* was strength of the normal interaction in the contacted nodes known after the previous step:(8)$$F_{N}=\sigma_{n}\cdot L_{x}\cdot t,$$
where *σ_n_* was normal stresses in the contacted nodes, *L_x_* was length of the contacted sample elements. Displacement of all nodes of the hard counterpart due to tangential load was equal to zero.

The boundary conditions for a node displacement along the *x*- and *y*-axes were set in all contacted sample nodes on the ‘BC’ boundary to avoid intersection of the surfaces and enable them to slide along the ‘EGF’ rigid counterpart surface [28]. 

For example, if the ‘*k*’ node (Figure 2) moved along the ‘lm’ boundary, the condition was set that prohibits movement at each of the contacted nodes:(9)$$u_{k}=v_{k}\cdot \cot a,$$
where *U* was movement of the node along the *x*-axis; α was the inclination angle of the ‘lm’ counterpart contact border to the *x*-axis, which was calculated for the ‘k’ node as:(10)$$\mathrm{ctg}a=(x_{m}-x_{l})/\left(y_{m}-y_{l}\right),$$

The corresponded functional derivatives were set as follows:(11)$$\frac{\partial \Pi }{\partial v_{k}}=\frac{\partial \Pi }{\partial v_{k}}+\frac{\partial \Pi }{\partial u_{k}}\cdot \frac{\partial u_{k}}{\partial v_{k}}=\frac{\partial \Pi }{\partial v_{k}}+\frac{\partial \Pi }{\partial u_{k}}\cdot \mathrm{ctg}a,$$

Some changes in the global stiffness matrix, similarly to Formula (3), are caused by this condition. The ‘*U_k_*’ rows and columns divided by tg α were added to the ‘*V_k_*’ corresponding rows and columns. Then, the equation corresponded to the ‘*U_k_*’ rows and columns of the stiffness matrix should be excluded from the system of the equations. However, the equation was left using the method of accounting for boundary conditions in terms of displacements [25] to preserve the stiffness matrix symmetry: the diagonal element in the row corresponding to ‘*U_k_*’ was multiplied by a “large” number (for example, 10^6^), and the free term was zeroed. After solving the system of the equations, ‘*U_k_*’ movement was determined by Formula (4).

The stress and strain distribution was determined based on the results of stress–strain analysis. The fracture criteria (the maximum normal and tangential stresses; the stress and strain intensity criterion corresponding to the material tensile strength values) were checked at every step. When one of them was fulfilled, the failure was concluded and material fragment was removed. As a result, these elements were taken out from the calculation process. After that, a new contact boundary was formed and the finite element mesh was rebuilt. In addition, all values of the stress–strain state parameters in the nodes and the elements of the new mesh obtained at the previous step (including the properties of the materials) were changed using the linear interpolation.

## 3. Simulation Results

Simulation results were obtained under the normal and shear loads at the ‘ABCD’ calculation domain (Figure 3b). A uniform material (UHMWPE) and a region loaded with carbon fibers of different lengths (inclusions) were analyzed. The normal load of 140 N was used by analogy with experimental studies of the UHMWPE samples (see results below). It was shown that, as the load increased, the shear deformation in the subsurface layer developed more intensity. This was the reason why the calculations at the highest load were used in the experimental investigations. The loading degree was 5% by volume (Figure 3) that corresponded to 10% by weight (since density of the carbon fibers was 1.800 g/cm^3^, which is about twice density of UHMWPE equal to 0.934 g/cm^3^). The inclusions had a rectangular shape; their location in the computational domain was set using a random number generator. The latter determined the position and the inclination angle. The ideal contact conditions between the fibers and the matrix were taken.

For the numerical simulation in the contact of the fibers and the polymer matrix, the ideal contact conditions were used. For this reason, the contribution of the fibers was determined identically both for tension and for compression. The authors understand that the main purpose of reinforcing fibers was increasing strength in the direction parallel to the reinforcement. In doing so, if the aspect ratio is small, the fibers act more like particles. However, their main function was to suppress the development of shear deformation in the subsurface layer.

The following properties of the uniform material (matrix) were used for the calculations: elastic modulus was 711 MPa; yield strength was 23.9 MPa; tensile strength was 37.5 MPa; elongation at break was 480%; the friction coefficient was 0.12. The elastic modulus of the filler (carbon fibers) was 50 GPa. The counterpart diameter of the GCr15 bearing steel was 60 mm. The simulations of UHMWPE under loading were performed taking into account its nonlinear behavior. The ‘BC’ initial contact surface of the samples (Figure 3b) was smooth.

Surfaces of the strain intensities and the contours of the UHMWPE samples showing their change with increasing the counterpart rotation number are presented in Figure 4. Regions corresponding to the strain intensity exceeding 10% are highlighted in red (hereinafter). It is seen that, with increasing tribological loading time, sample wear became more severe. Therefore, the size of the regions with a strain exceeding 10% was increased both along the contact surface and deep into the sample. 

As the worn out material was removed, surface protrusions and cavities had been formed. The maximum deformation peaks corresponded to such areas of a subsurface layer at the tribological contact. This comes from the fact that the maximum tangential stresses during the contact interaction of the cylinder and the elastic half-space under the normal compressive load had been equal to 0.304 *σ_max_*, while their depth had been 0.786 *a* (where *a* had been half of the contact area length) [30,31]. 

A detailed analysis of the strain tensor component distributions (Figure 5) allowed one to identify alternation of the areas having tensile or compressive stresses due to formation of protrusions and cavities on the contact surface. Compressive and tensile stresses (and, accordingly, strains) along the *x*-axis resulted from compression of the protrusions. In addition, tensile stresses had been formed since tangential load had acted on the protrusion peaks. In doing so, alternation of such areas occurred. Deformations due to both tensile and compressive stresses along the *x*-axis were approximately 20%. The main reason is related to the action of the ε*_xy_* component (up to 40% deformation). The maximum deformations in the *y*-axis direction under compressive load reached 16%, while they did not exceed 5% due to tensile load. The maximum tensile stresses under compressive load along the *z*-axis were developed. The latter caused material deformation up to 5%. Thus, the deformation resulted from alternated compression and tension of the protrusion peaks on the tribological contact surfaces.

In the case of tribological loading of the samples filled with 5% (vol.) inclusions (Figure 6b), the maximum strains were predominantly developed in the upper layer of the computational domain. No inclusions were there (Figure 6). In doing so, the strain distribution varied nonuniformly along the surface. However, the deformations had not extended in depth, as in the case of neat UHMWPE. This took place for the inclusions of different lengths.

Thus, it can be concluded from the above examples that the UHMWPE samples without any inclusions had been deformed to a greater depth, primarily due to lower elastic modulus. The maximum deformations were propagated deeper than in the samples with the inclusions. 

The same results were obtained under the normal load of 60 N, but the values of deformation, stress gradients, and the deformed layer thickness were lower.

It has been shown that loading with reinforcing fibers had suppressed the elastic and plastic deformations in the subsurface layer. Therefore, wear resistance improvement by loading with reinforcing fibers depends on the suppressing plastic flow of the polymer subsurface contact layers in the sliding direction under tangential load. The results of the experimental verification of these data are given below.

## 4. Material and Methods of Experimental Studies

The “Ticona GUR-2122” UHMWPE powder (Celanese Corporation, Irving, Texas, USA) was used to fabricate samples. Its molecular weight was 4.5 million; particle size was 5–15 µm; particles were weakly agglomerated into aggregates with a size of 120–150 µm. Data on fibrous fillers are presented in Table 1. The “Olenten” High–Density PolyEthylene grafted with Styrene Maleic Anhydride (HDPE-g-SMA) grafted high-density polyethylene was loaded as a compatibilizer (New Polymer Technologies LLC, Moscow, Russia). In the initial state, it was purchased in the form of granules 2–3 μm in size. Then, the polyethylene was mechanically milled using a “Rondol” drum grinder to a particle size of ~525 μm.

Mixing of the UHMWPE polymer binder powders and fillers was done using an “MP/0.5x4” planetary ball mill (Tekhnocenter LLC, Rybinsk, Russia). The components were preliminary dispersed using a PSB-Gals 1335–05 ultrasonic cleaner (PSB-Gals Ultrasonic equipment center, Moscow, Russia).

Bulk preforms of polymer composites were fabricated by hot pressing of two-component powder mixtures at a pressure of 10 MPa and a temperature of 200 °C using a laboratory setup based on a “MS-500” hydraulic press (NPK TekhMash LLC, Moscow, Russia). The setup was equipped with an open-loop ring furnace with digital temperature control (ITM LLC, Tomsk, Russia). After holding under pressure, the preforms were cooled without unloading for 30 min. Cooling rate was 5 °C/min.

Tensile properties of the “dog-bone” shaped UHMWPE-based samples were measured using an “Instron 5582” electromechanical testing machine (Instron, Norwood, Massachusetts, USA). The number of samples of each type was at least four.

“Pin-on-disk” dry sliding friction tests were performed to determine friction coefficients using a CSEM CH-2000 tribometer (CSEM, Neuchâtel, Switzerland). Load was 5 N; contact pressure *P_max_* was 31.8 MPa; sliding speed was 0.3 m/s. A ball-shaped counterpart 6 mm in diameter was made of the GCr15 bearing steel. 

Wear resistance was evaluated according to the “block-on-ring” scheme using a “2070 SMT-1” friction testing machine (Tochpribor Production Association, Ivanovo, Russia). Load on the samples was 60 and 140 N (contact pressure *P_max_* was 9.7 MPa and 32.4 MPa); sliding speed was 0.3 m/s and 0.5 m/s. A counterpart was made of the outer ring of a commercial bear. It had a disk shape with a diameter of 35 mm and a width of 11 mm. Counterpart surface roughness was 0.20–0.25 μm. Counterpart temperature was measured using a CEM DT-820 non-contact InfraRed (IR) thermometer (Shenzhen Everbest Machinery Industry Co., Ltd., Shenzhen, China).

Wear rate was determined by measuring width and depth of the wear track according to stylus profilometry, followed by multiplication by its length. The wear rate calculation was done according to a widespread method taking into account the load and distance values:Wear rate = volume loss (mm^3^)/sliding distance (m).

The wear track profiles were determined using the data on at least 10 tracks. Then, the wear rate calculation was carried out on the basis of the experimental test data on at least four samples of each type. Mathematical statistics methods were used for the experimental results processing.

Surface topography of the wear tracks was studied using a Neophot 2 optical microscope (Carl Zeiss, Oberkochen, Germany) equipped with a Canon EOS 550D digital camera (Canon Inc., Tokyo, Japan), and an Alpha-Step IQ contact profiler (KLA-Tencor, Milpitas, California, USA).

The cleaved surfaces of the notched specimens mechanically fractured after exposure in liquid nitrogen were used for permolecular structure studies. A LEO EVO 50 scanning electron microscope (Carl Zeiss, Oberkochen, Germany) was employed (accelerating voltage was 20 kV). Crystallinity was determined using a SDT Q600 combined analyzer (TA Instruments, New Castle, Delaware, USA).

## 5. Experimental Results and Discussion

### 5.1. Wear of Neat UHMWPE under Various Load–Sliding Speed Conditions of Tribological Loading

The dependence of neat UHMWPE wear rate versus applied load at a fixed counterpart speed of 0.3 m/s is shown in Figure 7. It is seen that it increased nonlinearly as applied load rose. Specific pressure on the samples also decreased nonlinearly. An increase in load from 30 to 100 N resulted in growing wear rate by 2.3 times, while, with an enlargement in load from 100 to 140 N, wear rate grew by about six times. In this case, specific pressure decreasing was not in the direct proportion to the applied load increase. This, highly likely, was associated with the intensive material removal with its growth. 

Such dynamics in wear rate was accompanied by an increase in the nominal area of the sample–counterpart tribological contact. The latter was determined through the wear track area; as it expands with wear, the contact area increased as well. New surface areas, whose material had not yet been run-in, had worn faster than the surface material of the previously formed wear track. These results were consistent with the data of [32]. It was shown that an increase in the contact surface areas of the neat UHMWPE friction surface samples had been accompanied by an increase in wear rate at the same specific pressure. This resulted from polymer molecule reorientation in the run-in tribological surface layer in the sliding direction. At the same time, wear rate increased both in non-run-in and slightly run-in material. 

In addition, an increase in the nominal tribological contact area due to applied load growth gave rise to counterpart heating (Figure 8). Heating rate gradually decreased with wear track area increasing. It is most likely that this resulted from more intense material removing from the tribological contact surface, as well as the material plastic flow towards the counterpart sliding direction. It should be noted that wear rate and counterpart temperature changed in the opposite manner as the tribological contact area increased.

Note that UHMWPE wear track surface roughness remained almost constant at various loads, and was equal to that of the counterpart surface (0.20 ± 0.05 μm). This indicated that the wear process did not qualitatively change over the entire range of the tribological test load parameters at a speed of 0.3 m/s. These data were consistent with the papers [32,33,34], where an increase in the tribological contact area accompanied by wear rate growing was shown.

During the dry sliding friction tests at a fixed load of 60 N, wear rate of the neat UHMWPE increased by more than 10% with more than double speed growth (Figure 9). At the same time, specific pressure to the tribological contact area decreased linearly. The reason is determined by increase in the wear track area due to material removing. Thus, a change in sliding speed was responsible for an increase in wear rate to a much lesser extent compared to variation in the applied load (Figure 7).

At a high sliding speed (0.7 m/s), a small tribological contact area (26.6 mm^2^) was formed, while the counterpart was heated only to a temperature of 39 °C. For comparison, about the same counterpart temperature of 37.2 °C was under the maximum load (140 N) in the studied range when the nominal tribological contact area (95.6 mm^2^) was four times higher (Figure 9). Thus, an increase in sliding speed was the expected reason of a linear increase in counterpart heating temperature.

It should be noted that the measured counterpart temperature values was rather a qualitative than a quantitative indicator of the wear process. For sure, temperature could reach significantly higher values in the contact spots. However, according to the authors, the data of the non-contact IR thermometry of the steel counterpart surface might be correctly employed for quantitative comparing and interpretation. Summarized dependence of UHMWPE wear rate on the ‘P·V’ load–sliding speed parameter is shown in Figure 10. 

The obtained results generally coincided well with the published data on correlation between sliding speed and UHMWPE wear rate [34,35]. In these papers, an increase in sliding speed did not give rise to a significant growth of UHMWPE wear rate.

Unlike many published results of the UHMWPE tribological tests (for example, [36,37]), the “pin-on-disk” scheme was not used in this work. This is governed by the fact that specific pressure on the friction surface exceeded the material yield strength. The value of contact area between a UHMWPE sample and the steel counterpart exerted a significant impact on wear rate for the used “block-on-ring” scheme due to wider distributions of the applied load. In addition to the above-mentioned effect of temperature on the polymer plasticization in the tribological contact area, the actual contact area between the sliding surfaces had increased significantly with load growing. For this reason, the counterpart had impacted on a larger area, resulted in intensive wear of the non-run-in regions. In contrast to varying sliding speeds, an increase in load exerted a more significant effect on neat UHMWPE wear rate during the tribological tests. The listed factors are shown in Figure 11. It is seen that fragments of the surface layer in separate tribological contact areas moved along the sliding direction under shear load transmitted from the rotated counterpart. A similar effect had been especially pronounced when both high applied load and sliding speed occurred simultaneously.

Material removed from the surface layer under “lighter” load–sliding speed conditions of tribological loading was less pronounced (Figure 11a,f). At the same time, the SEM micrographs of the subsurface wear layer after the end of the tribological tests (Figure 11f) illustrated signs of the pronounced irreversible shearing deformation there. Fragments of the formed “layered” structure in the subsurface layer were oriented in the sliding direction. Note that the neat UHMWPE permolecular structure represented a set of spherulites having the fibril orientations along the radii from their centers [38].

A similar layered structure both near the surface and at a certain distance away was formed in the entire range of the applied loads (60–140 N). Under the lowest load (60 N), the shear induced layering (“layered mesostructure”) was maximally expressed near the surface with gradual decreasing towards the bulk material (Figure 11f). Depth of such an irreversibly deformed layer was 100–150 µm. Two structural levels of the formed layered mesostructures were found with an increase in both speed and load (to a greater extent). One was near the surface; the second was at a depth of more than 100 µm. This effect was most pronounced under the maximum load of 140 N (Figure 11g,h).

With increasing the sliding speed up to 0.5 m/s, the layered mesostructure was formed, but with a smaller size of the structural elements (Figure 11h). It can also be described as a “blocked” one. The thickness of the layers had decreased to ~1.3 μm. It was six times less than at a low sliding speed (0.3 m/s). The reason of the layered structure formation was shear load transmission from the rotated counterpart. As a result, fragments of the wear track surface layer had moved along the sliding direction and the material had been deformed layer by layer. The obtained results were consistent with the proposed model [39], which reflected the deformation of different sample areas in depth during the UHMWPE wear.

The results of the counterpart temperature measurement showed that wear rate and temperature were increased rapidly with ‘P·V’ growing. This was accompanied by an increase in the wear track area (Figure 12). An increase in counterpart temperature gave rise to “wrinkles” formation on the wear track surface (Figure 11). The latter is responsible for its roughness enlarging (Figure 13). However, a significant increase in roughness took place at the maximum studied ‘P·V’ (70 N·m/s) only. The intense surface layer flows had been accompanied by formation of the wrinkles (Figure 11d) due to frictional heating.

Based on the obtained results, further studies were aimed at UHMWPE wear resistance improving by loading with carbon fibers of different sizes. Their loading should prevent the shear deformation and suppress the plastic flow in the subsurface tribological contact layer (especially at high P·V) due to the reinforcement effect.

### 5.2. UHMWPE-Based Composites Loaded with Carbon Fibers of Different Sizes

It was shown in [40] that carbon fibers of various sizes (from nano- to millimeter) were able to simultaneously increase the mechanical and tribological properties of the UHMWPE-based composites. In this case, just 0.5 wt.% of carbon nanofibers effectively improved UHMWPE wear resistance [41] while the strength properties were constant. Loading of at least 10 wt.% of the micro- and millimeter-sized carbon fibers provided an increase in the mechanical characteristics of UHMWPE composites [42]. 

The mechanical properties of the UHMWPE-based composites loaded with the carbon fibers of different sizes are shown in Table 2. It is seen that density and Shore D hardness of the composites are increased slightly compared with neat UHMWPE. Elastic modulus of the composites grew sharply with increasing length of the carbon fibers. The properties of the multicomponent UHMWPE-based composite simultaneously loaded with the carbon (nano)fibers, the chopped carbon fibers (a few millimeters long) as well as the grafted HDPE-g-SMA as a compatibilizer (it was used to increase adhesion of the fillers to a non-polar ultra-high molecular weight matrix) are also shown in Table 2.

A 2.3-fold increase in elastic modulus (from 711 MPa for neat UHMWPE up to 1672 MPa for the CCF (hereinafter abbreviations according to Table 1) reinforced composite) was found. Dynamics of changes in yield strength of the composites were similar. Its value has increased from 21.6 MPa for neat UHMWPE up to 33.5 MPa for the CCF reinforced composite. At the same time, crystallinity of the composites has decreased. The mechanical properties of the composite and their ability to absorb mechanical energy under impact loading have slightly decreased compared with neat UHMWPE.

The permolecular structure of the composites loaded with the carbon fibers of various sizes is illustrated by Figure 14a–d. The CNF reinforced composite (Figure 14a) possessed a spherulitic permolecular structure with half the size of the elements compared with the neat UHMWPE. The CNF was distributed predominantly along the spherulite boundaries. However, the MCF was fairly evenly distributed throughout the matrix (Figure 14,b). In comparison with neat UHMWPE, the spherulite sizes have decreased by 2.5–3.0 times. CCF was less evenly distributed throughout UHMWPE matrix (Figure 14c) due to their great lengths and the dry mixing method used. The permolecular structure was fairly uniform in the multicomponent composite (Figure 14d).

The tribological properties of the UHMWPE-based composites loaded with the carbon fibers with various aspect ratios were then investigated. The results of the friction coefficient measurements are presented in Figure 15. It is seen that loading with the CNF resulted in a halving friction coefficient compared with neat UHMWPE due to the solid lubricant effect of the nanofiller [43]. When loading 10% MCF, the friction coefficient initially significantly increased from 0.06 up to 0.15, and then remained stable until the end of the tribological tests. At loading with 10% CCF, the friction coefficient decreased by 30%, compared with neat UHMWPE. Note that it maintained a stable value throughout the entire tribological loading time. According to the authors, a decrease in the friction coefficient after loading with CCF is governed by reducing the relative content of the carbon fibers on the friction surface. This resulted from their long length and a less “retarding” effect on the metal counterpart. The friction coefficient for the multicomponent composite was equal to that for the nanocomposite (columns 2 and 5).

The dependence of wear rate versus the content of the carbon fibers of different sizes for various tribological test parameters is shown in Figure 16. One can conclude from the presented data that UHMWPE loading with the CNF ensured best efficiency in the conditions of low load and low sliding speed, as well as high load and high sliding speed (P·V = 18 and 70 N·m/s). In these cases, wear resistance of the composites was more than doubled compared with neat UHMWPE. Wear resistance of the composites loaded with MCF and CCF was almost identical. Their loading resulted in the highest efficiency in the conditions of high load and high sliding speed (P·V = 42 and 70 N·m/s). In doing so, wear resistance had increased from two to three times in comparison with neat UHMWPE. However, the steel counterpart experienced an abrasive wear.

The obtained data on wear resistance of the composites loaded with the carbon fibers of various sizes were in good agreement with topography of the wear track surfaces shown in Figure 17. It is seen that loading with the CNF had not favored formation of the wrinkles on the friction surface and the shear deformation structures in the subsurface wear track layer (Figure 17a,e). MCF (Figure 17b,f) and CCF (Figure 17c,g) had reinforced the subsurface wear layer provided redistribution of the load being transmitted from the sliding counterpart. The loading with CNF and CCF (nano- and millimeter-sized) had simultaneously ensured reinforcement of the subsurface layer and the solid lubricant effect (Figure 17d,h).

The optical images of the counterpart surfaces after the end of “pin-on-disk” tribological tests are shown in Figure 17i–l. It is seen that loading with CNF resulted in minimal counterpart wear (Figure 17l). A significant amount of the fibers was located and protruded above the sliding surface of the MCF-containing polymer composite due to their small length (200 μm). In doing so, microabrasive wear and an increase in the friction coefficient had taken place due to their contact with the counterpart (Figure 15). CCF, having a large length (2 mm), protruded above the sliding surface to a lesser extent. This resulted in a minimal counterpart wear (Figure 17l).

Thus, loading with the micro- and millimeter-sized reinforcing fibers had prevented formation of the wrinkles on the wear track surface and had reduced deformation as well as fatigue wearing of the subsurface layer. However, the carbon fibers had experienced failure due the action of the compressive and shear forces transmitted from the steel counterpart. Therefore, the CCF was beneficial in terms of improving the mechanical properties of the composite, while the CNF were valuable as a solid lubricant (the UHMWPE + 10% HDPE-g-SMA + 2% CCF + 0.5% CNF multicomponent composite). It should be especially noted that temperature on the friction surfaces also decreased in the composites loaded with the nano- (CNF) and millimeter-sized (CCF) carbon fibers (Figure 18) both under “light” (P·V = 18 N·m/s) and “heavy” conditions of the tribological testing (P·V = 70 N·m/s). Most likely, this effect comes from a higher thermal conductivity coefficient of the filler (more than 3000 W/m K) compared with neat UHMWPE.

Thus, under the “moderate” tribological test conditions (60 N, 0.3m/s), the CNF was the most efficient filler (with a loading degree up to 0.5%). The latter acted as a solid lubricant. This resulted in increasing wear resistance by 2.7 times. Under the “heavy” testing conditions (140 N, 0.5 m/s), the CCF with a length of 2 mm and the optimal loading degree of 10% were more efficient. They perceived the action of compressive and shear loads from the counterpart and protected the polymer composite contact surface from intense wear. Eventually, wear resistance doubled, other mechanical properties also improved. Note that simultaneous loading of the CNF as a solid lubricant and the CCF as reinforcing carbon fibers had provided the prescribed mechanical and tribological properties in the entire investigated range of the P·V load–sliding speed conditions of tribological loading.

## 6. Remarks and Discussion

The main goal of the analytical section of the manuscript was to show that the development of intense shear deformations under the wear track could be suppressed by filling the polymer with the fibers. At the same time, the authors did not aimed at achieving complete quantitative correspondence between the calculated and experimental data, since only qualitative conclusions could be made on the basis of the SEM observation. However, since the development of shear deformation was found experimentally, and its nature correlated with the load–speed parameters of the tribological tests, the authors made an attempt to theoretically justify the effect of the fibers on an increase in wear resistance. This was carried out not in the classical sense (a change in the interaction of the metal counterpart with the polymer), but via a change in the subsurface layer stress–strain distribution.

Another justification for the relevance of studying the wear mechanism of UHMWPE loaded with carbon fibers is the following. The main method to improve UHMWPE wear resistance was its loading with micro- and nanoparticles. Being tightly pressed into the polymer matrix, in the absence of interfacial adhesion, the filler particles enabled increasing this material property by several times. Meanwhile, features of increasing wear resistance due to the load redistribution from the polymer to the harder inclusions were discussed by many authors [44,45]. Loading with the fibers improved mechanical properties but was rarely discussed as a way to increase wear resistance. As shown above, UHMWPE wear rate was significantly determined by its elastic and inelastic deformations under the wear track. For this reason, the authors performed the detailed theoretical analysis of the effect of loading with the fibers on the stress–strain distribution in the subsurface layer of the composite. This enabled interpreting the results of the experimental tribological studies more correctly. 

The authors would also like to comment on the terms used. The phrase “moderate” loading conditions meant the ones that caused a low wear rate (comparable to that traditionally used in similar published studies), as well as not being accompanied by a noticeable change in counterpart temperature. In this case, wear developed mainly by the fatigue mechanism, and there were no micro-grooves or deformations on the friction surface. Increases in load or speed under tribological loading gave rise to enlarging temperature and wear rate. 

The same is related to practical application of the composites. The studies were devoted to the development of the wear-resistant polymer-matrix composites for operation in the dry friction conditions and in the wide temperature range (from −80 to +80 °C). Loading with inexpensive reinforcing fillers was aimed to improve their mechanical properties as well as wear resistance. Promising areas of their practical use are, for example, the manufacture of lining plates for construction and marine equipment, chippers for transport infrastructure, etc.

Two tribological test schemes were employed in the study, i.e., "pin-on-disk" and "block-on-ring". The available CSEM CH2000 tribometer made it possible to measure the friction coefficients. However, high specific pressures took place in the tribological contact. It was shown that this scheme of UHMWPE tribological testing was not always sensitive enough to changes in its structure and mechanical properties. At the same time, the friction testing machine that implements the “block-on-ring” test scheme did not enable measuring the friction coefficients. However, the load–speed parameters of the tribological tests could vary widely. In addition, it was possible to analyze in details the changes on the friction surfaces, as well as the subsurface layer structure due to larger samples and lower specific pressure in the tribological contact.

At various parts of the manuscript, the improvement of the tribological properties is discussed. It is a simplified interpretation since in some applications one should “increase” frictional resistance, while, in lubrication applications, it is necessary to decrease frictional resistance. The key goal of the work was to increase wear resistance of the polymer composites. The authors analyzed a change in the friction coefficients only in terms of interpreting the observed effects. The task of frictional resistance reducing was not considered in the manuscript.

Finally, let us again comment on the temperature measurements. The tribological tests started at room temperature. As was noted above, the temperature control method used by the authors was rather approximate. However, the following temperature analysis principle was used. Almost no temperature changes occurred at the “moderate” loading conditions, and the authors did not even discuss its fluctuations within 2–3 degrees, referring this to a possible dispersion or errors. Therefore, the authors considered tens of degrees as a significant change in temperature (it was taken into account when interpreting the results of changes in wear resistance). In this case, the signs of strain intensification on the surface of the wear tracks due to visually observed frictional heating were registered. Currently, an advanced laboratory device is being designed for a more accurate and reliable temperature measurement in tribological contacts.

## 7. Conclusions

A structural analysis of the surfaces and the subsurface layers of UHMWPE as well as the UHMWPE-based composites loaded with the carbon fibers of various (nano-, micro-, millimeter) size in a wide range P·V tribological loading conditions was performed.

The possibility to control the wear processes of the tribological contact polymer parts by loading with reinforcing inclusions that suppress the elastic and inelastic deformations was evaluated. For this purpose, the computer simulation (including parametric studies) was implemented to determine the parameters of the stress–strain state that influenced the dynamics of these processes.

An analysis of effects of carbon fiber sizes on structure formation of the subsurface tribological contact layer and wear resistance of antifriction composites were carried out. On the basis of the computer simulation results, the relationship between the subsurface layer properties and the tribological characteristics of UHMWPE and the UHMWPE-based composites was presented.

It was shown that, under the “moderate” conditions of the tribological tests (60 N, 0.3 m/s), the carbon nanofibers was the most efficient filler (with a loading degree up to 0.5 wt.%). In doing so, they acted as a solid lubricant. As a result, wear resistance increased to 2.7 times. Under the “heavy” test conditions (140 N, 0.5 m/s), the chopped carbon fibers with a length of 2 mm and the optimal loading degree of 10 wt.% were more efficient. They perceived the action of compressive and shear loads from the counterpart and protected the tribological contact surface from intense wear. In doing so, wear resistance doubled, and other mechanical properties also improved.

It was found that simultaneous loading of UHMWPE with CNF as a solid lubricant and LCF as reinforcing carbon fibers provided the prescribed mechanical and tribological properties in the entire investigated range of the load-sliding speed conditions of tribological testing.

## Figures and Tables

**Figure 1 materials-13-00338-f001:**
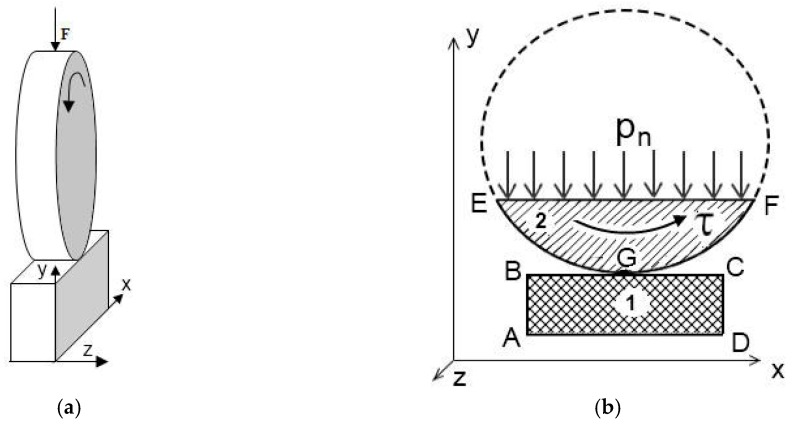
Schemes of the mechanical loading during friction (**a**) and computational domain (**b**).

**Figure 2 materials-13-00338-f002:**
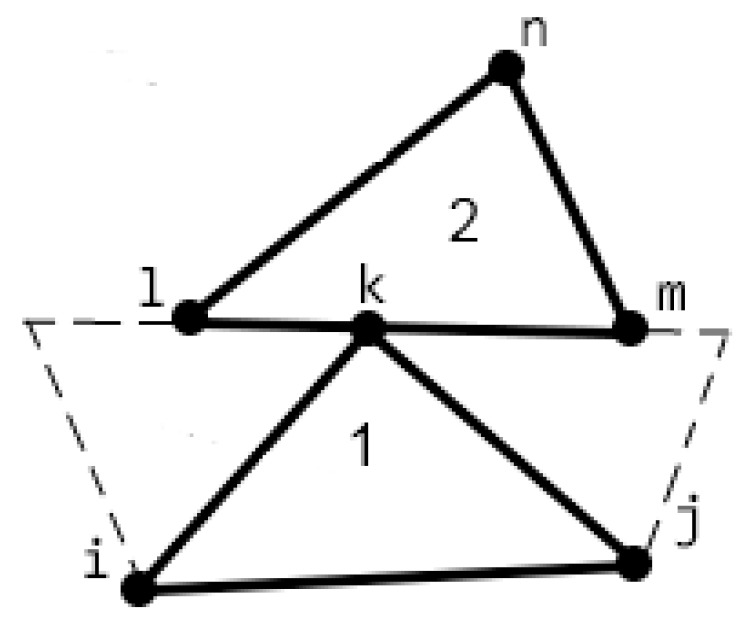
Finite element contact scheme.

**Figure 3 materials-13-00338-f003:**
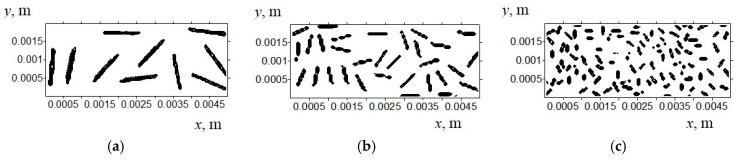
Computational domain for UHMWPE filled with carbon fibers with a length of: (**a**) 1000 µm; (**b**) 500 µm; (**c**) 200 µm.

**Figure 4 materials-13-00338-f004:**
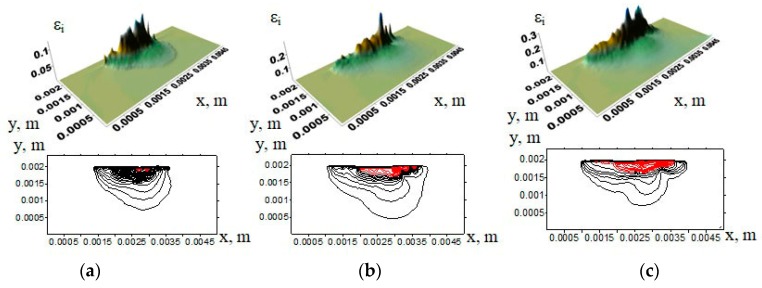
Strain distribution surfaces and their corresponding contours in UHMWPE samples under applying normal and tangential loads at maximum strain intensity: (**a**) 10%; (**b**) 20%; (**c**) 30%.

**Figure 5 materials-13-00338-f005:**
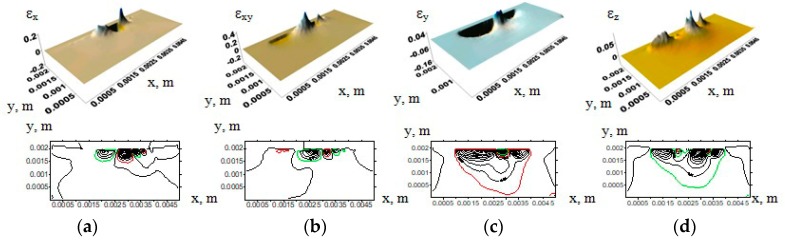
Strain tensor component distribution surfaces and their corresponding contours in uniform (neat) samples under normal and tangential loads: (**a**) ε_x_; (**b**) ε_xy_; (**c**) ε_y_; (**d**) ε_z_.

**Figure 6 materials-13-00338-f006:**
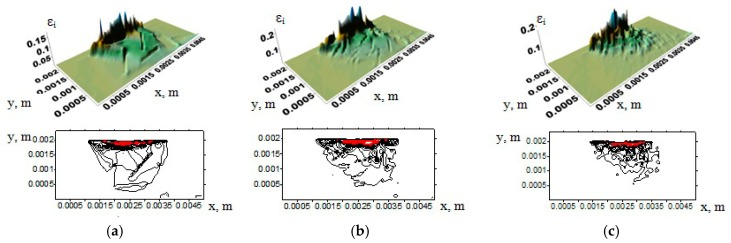
Strain distribution surfaces and their corresponding contours in samples filled with inclusions with a length of: (**a**) 1000 µm; (**b**) 500 µm; (**c**) 200 µm.

**Figure 7 materials-13-00338-f007:**
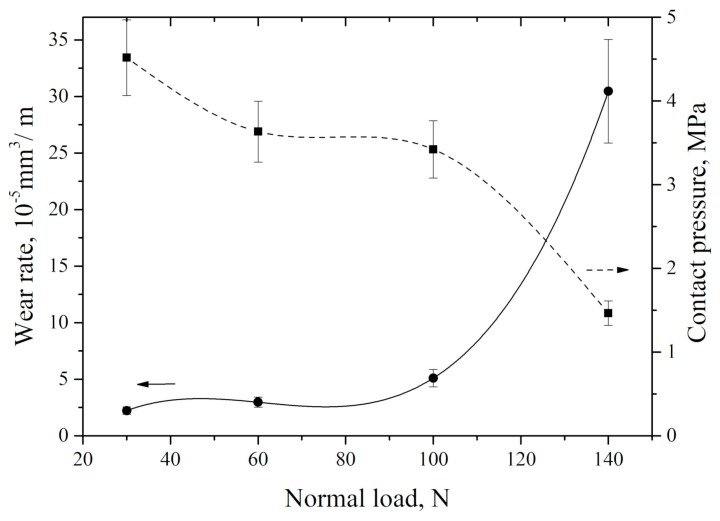
Neat UHMWPE wear rate vs. specific pressure on the wear track at different loads.

**Figure 8 materials-13-00338-f008:**
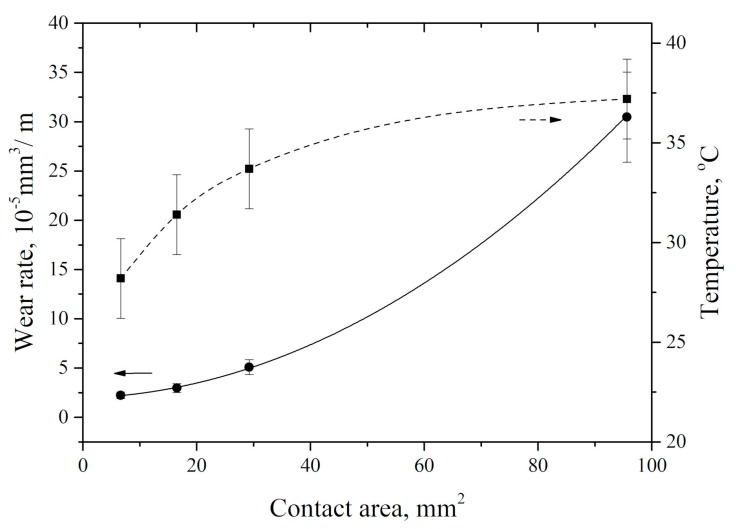
Counterpart temperature and wear rate vs. nominal sample–counterpart contact area.

**Figure 9 materials-13-00338-f009:**
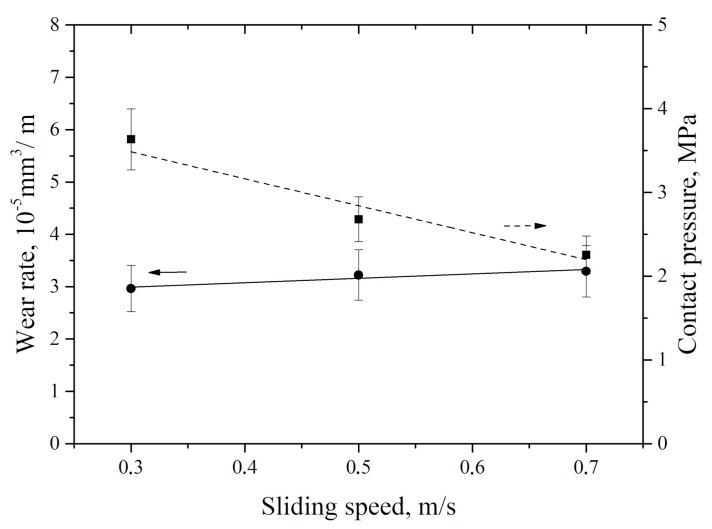
UHMWPE wear rate vs. sliding speed.

**Figure 10 materials-13-00338-f010:**
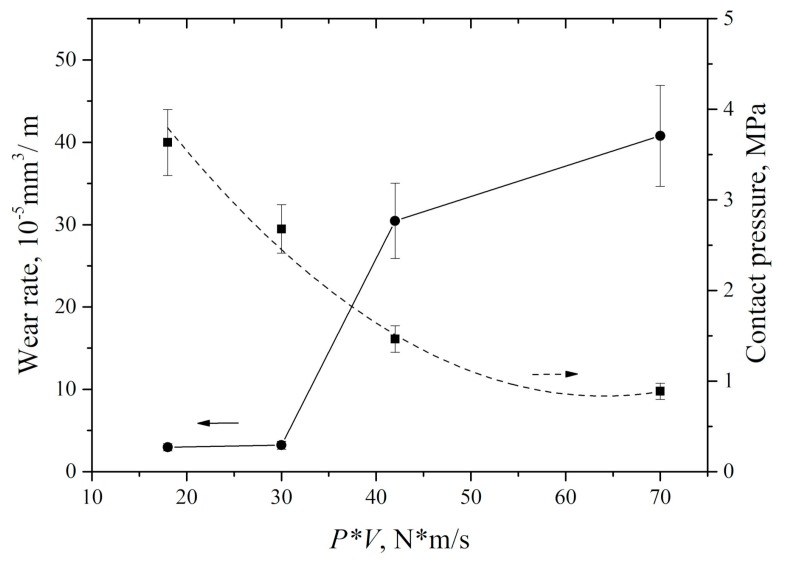
UHMWPE wear rate and specific pressure on the wear track vs. *P*·*V* tribological loading parameter.

**Figure 11 materials-13-00338-f011:**
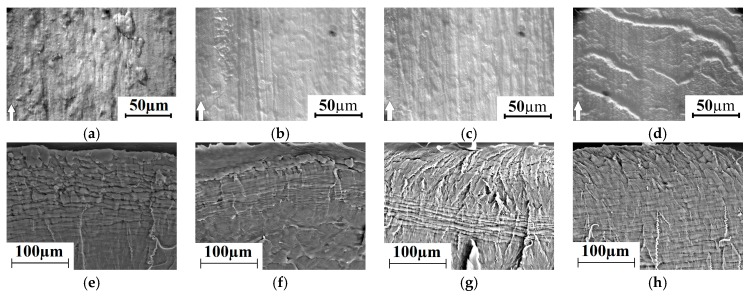
SEM-micrographs of the UHMWPE wear surfaces and subsurface layers: (**a**,**e**)—P = 60 N, V = 0.3 m/s; (**b**,**f**)—P = 60 N, V = 0.5 m/s; (**c**,**g**)—P = 140 N, V = 0.3 m/s; (**d**,**h**)—P = 140 N, V = 0.5 m/s.

**Figure 12 materials-13-00338-f012:**
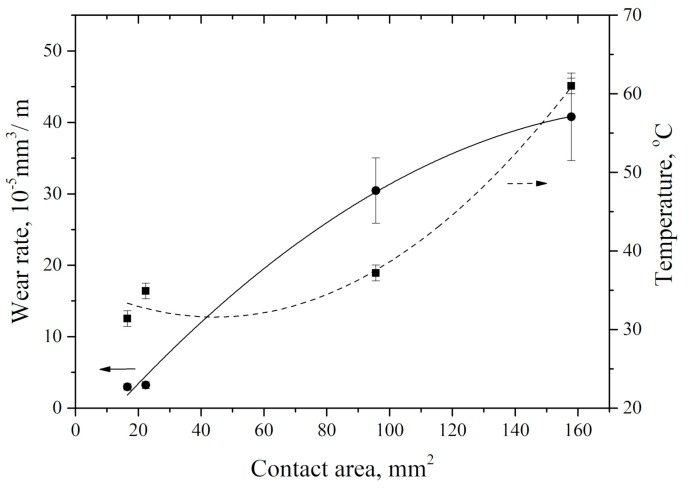
Counterpart temperature and wear rate vs. wear track area.

**Figure 13 materials-13-00338-f013:**
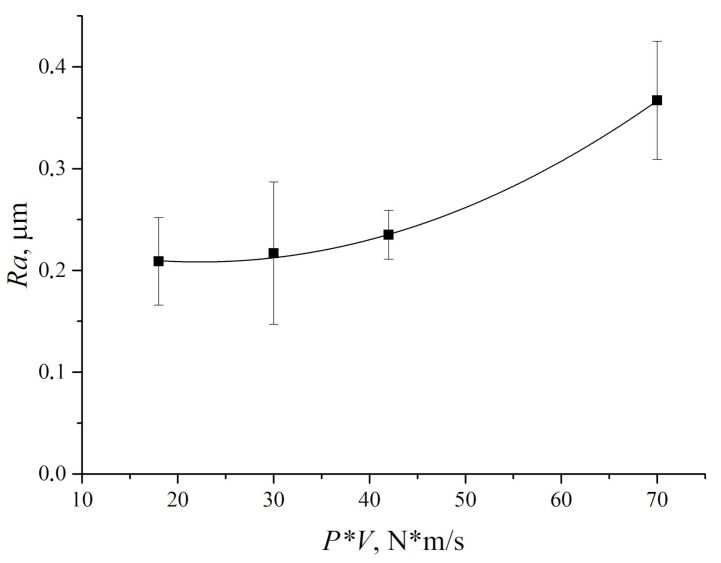
UHMWPE surface roughness vs. *P*·*V* tribological loading parameter.

**Figure 14 materials-13-00338-f014:**
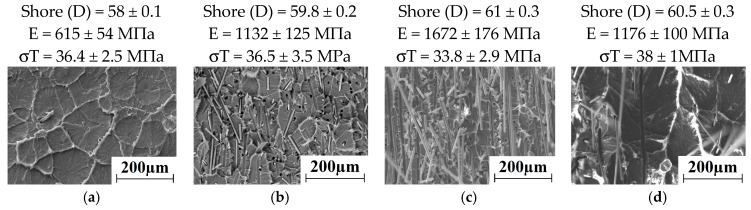
SEM-micrographs of the permolecular structure: (**a**) UHMWPE + 0.5% CNF; (**b**) UHMWPE + 10% CF; (**c**) UHMWPE + 10% LCF; (**d**) UHMWPE + 10% HDPE-g-SMA + 2% LCF + 0.5% CNF.

**Figure 15 materials-13-00338-f015:**
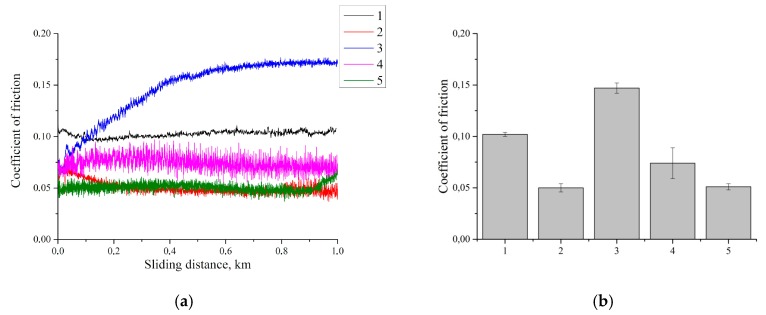
Change in friction coefficient during tribological tests (**a**) and its average values (**b**): 1—UHMWPE; 2—UHMWPE + 0.5% CNF; 3—UHMWPE + 10% CF; 4—UHMWPE + 10% LCF; 5—UHMWPE + 10% HDPE-g-SMA + 2% LCF + 0.5% CNF.

**Figure 16 materials-13-00338-f016:**
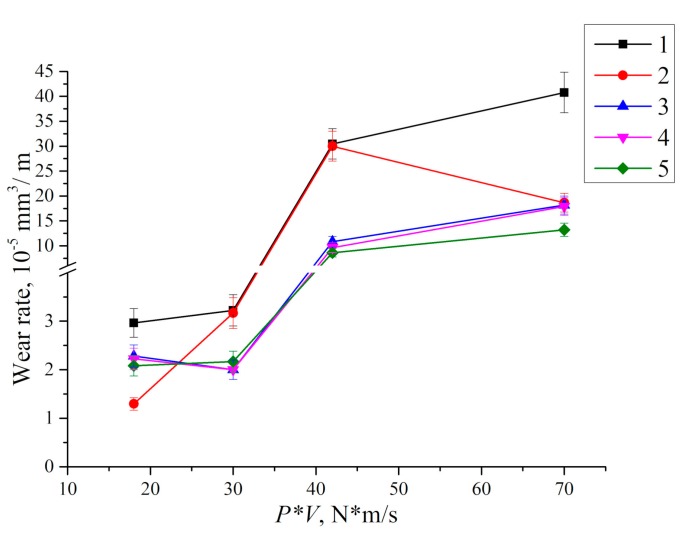
Wear rate vs. *P*·*V* tribological loading parameter: 1—UHMWPE; 2—UHMWPE + 0.5% CNF; 3—UHMWPE + 10% CF; 4—UHMWPE + 10% LCF; 5—UHMWPE + 10% HDPE-g-SMA + 2% LCF + 0.5% CNF.

**Figure 17 materials-13-00338-f017:**
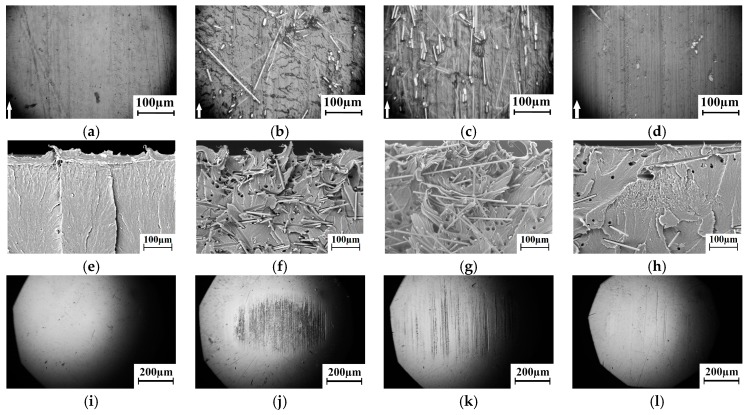
Optical images of the wear track surfaces and (**a**–**d**), the SEM-micrographs of the permolecular structure of the wear subsurface layer (**e**–**h**), and the counterpart surfaces (**i**–**l**) after tests at *P* = 140 N and *V*=0.5 m/s: (**i**) UHMWPE + 0.5% CNF; (**j**) UHMWPE + 10% CF; (**k**) UHMWPE + 10% LCF; (**l**) UHMWPE + 10% HDPE-g-SMA + 2% LCF + 0.5% CNF.

**Figure 18 materials-13-00338-f018:**
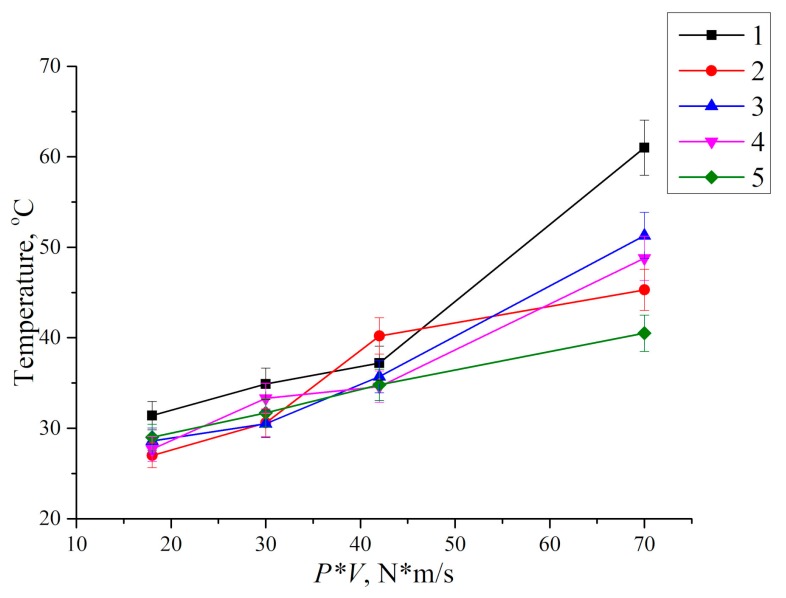
Counterpart temperature vs. *P*·*V* tribological loading parameter: 1—UHMWPE; 2—UHMWPE + 0.5% CNF; 3—UHMWPE + 10% CF; 4—UHMWPE + 10% LCF; 5—UHMWPE + 10% HDPE-g-SMA + 2% LCF + 0.5% CNF.

**Table 1 materials-13-00338-t001:** Fibrous fillers used for the composite fabrication.

Type	Mean Length, μm	Diameter, nm	Aspect Ratio	Brand	Manufacturer
CNF (Carbon NanoFibres)	2	60	33	“Taunit”	NanoTechCenter LLC, Tambov, Russia
CF (Milled Carbon Fibers, MCF)	200	6.000	33	“UMT”	UMATEX, Chelyabinsk, Russia
LCF (Chopped Carbon Fibers, CCF)	2000	6.000	333	“UMT”	UMATEX, Chelyabinsk, Russia

CNF—Carbon Nano Fibers; CF—Carbon Fibers; LCF—Long Carbon Fibers.

**Table 2 materials-13-00338-t002:** Mechanical properties of UHMWPE and UHMWPE-based composites.

Filler Composition, % (wt.)	Density (*ρ*), g/cm^3^	Shore (*D*) Hardness	Elastic Modulus (*G*), MPa	Yield Strength (σ_Y_), MPa	Tensile Strength (σ_T_), MPa	Elongation at Break (ε), %	Impact Toughness (*a*), kJ/m^2^	Crystallinity (χ), %
None	0.928	57.5 ± 0.1	711 ± 40	21.6 ±0.6	42.9 ± 3.1	485 ± 28	151 ± 6	56.5
0.5% CNF	0.933	58.0 ± 0.1	615 ± 54	22.3 ± 0.3	36.4 ± 2.5	398 ± 40	134 ± 5	51.1
10% CF	0.974	59.8 ± 0.2	1132 ± 125	27.2 ± 0.4	36.5 ± 3.5	374 ± 36	119 ± 7	40.8
10% LCF	0.970	61.0 ± 0.3	1672 ± 176	33.5 ± 1.6	33.8 ± 2.9	279 ± 29	122 ± 8	34.4
10% HDPE-g-SMA + 2% LCF + 0.5% CNF	0.950	60.5 ± 0.3	1176 ± 100	29.1 ± 0.9	38.1 ± 2.0	364 ± 28	120 ± 6	42.3

UHMWPE—UltraHigh Molecular Weight Polyethylene; CNF—Carbon Nano Fibers; CF—Carbon Fibers; LCF—Long Carbon Fibers; High–Density PolyEthylene grafted with Styrene Maleic Anhydride (HDPE-g-SMA).

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
