# Peer review of "Increasing Wear Resistance of UHMWPE by Loading Enforcing Carbon Fibers: Effect of Irreversible and Elastic Deformation, Friction Heating, and Filler Size"

_materials, 2020, doi:10.3390/ma13020338_

Round 1

Reviewer 1 Report

The link between analytical and experimental studies was not very clear. Please elaborate more on this issue. Please explain why a normal load of 140N was specifically chosen for the numerical study. Fibres’ contribution was in both compressive and tensile stresses? The exact mechanism was not described thoroughly in the manuscript. It was not very much clear how wear rate was evaluated. Please provide some more detailed descriptions. Changes of wear rate might also be, partly, due to scatter in the data because of experimental errors. How the reproducibility of the experimental results and associated inherent errors were eliminated from the measurements? At various parts of the manuscript the authors discussed on the improvement of the tribological properties. It should be more explicit what is meant by this as in some applications it is desirable to “increase” frictional resistance, while in applications such as lubrication, it is desirable to decrease frictional resistance. Introduction, line 16: Expression “To achieve it” should be “To achieve this”. Abstract, lines 17-19: Sentence should be corrected as “the effects…. were evaluated” Line 43: Please check and use preposition “the” when applicable (it is missing from many parts of the text. An example in line 43: “This expands the applications …” Line 258: The expression should be corrected as “after the end of the tribological tests”. Line 325: The expression should be corrected as “The results of friction coefficient measurements”.

Author Response

Dear Reviewer, greatest thanks for your efforts to improve the manuscript. please find enclosed some of our remarks. The corrections in the revised manuscrpit are marked with cyan.

The link between analytical and experimental studies was not very clear. Please elaborate more on this issue.

The main goal of the analytical section of the manuscript was to show that the development of intense shear deformations under the wear track could be suppressed by filling the polymer with fibers. At the same time, the authors did not want to achieve complete quantitative correspondence between the calculated and experimental data, since only qualitative conclusions could be made ob the basis of the SEM observation. However, since the development of shear deformation was found experimentally, and its nature depended on the load–speed parameters of the tribological tests, the authors made an attempt to theoretically justify the effect of the fibers on an increase in wear resistance not in the classical sense (a change in the interaction of the metal counterpart with the polymer), but via a change in the subsurface layer stress–strain distribution.

Please explain why a normal load of 140N was specifically chosen for the numerical study.

In the calculations, the normal load of 140 N was used by analogy with experimental studies of the UHMWPE samples. It was shown that the shear deformation in the subsurface layer developed as the load increased. This was the reason of the calculations at the highest load used in the experimental investigations.

Fibres’ contribution was in both compressive and tensile stresses? The exact mechanism was not described thoroughly in the manuscript.

In the contact of the fibers and the polymer matrix, ideal contact conditions were used for the numerical simulation. For this reason, the contribution of the fibers was determined identically both for tension and for compression. The authors understand that the main purpose of reinforcing fibers is to increase strength in the direction parallel to the reinforcement. Fibers act more like particles if the aspect ratio is small. However, their main function was to suppress the development of shear deformation in the subsurface layer.

It was not very much clear how wear rate was evaluated. Please provide some more detailed descriptions.

Wear rate was determined by measuring width and depth of the wear track according to contact profilometry, followed by multiplication by its length. The wear rate calculation was done according to the widespread methods taking into account the load and distance values.

Changes of wear rate might also be, partly, due to scatter in the data because of experimental errors. How the reproducibility of the experimental results and associated inherent errors were eliminated from the measurements?

The wear track profiles were determined using the data on at least 10 tracks. Then, the wear rate calculation was carried out on the basis of the experimental test data on at least four samples of each type. The experimental results were processed using the mathematical statistics methods.

At various parts of the manuscript the authors discussed on the improvement of the tribological properties. It should be more explicit what is meant by this as in some applications it is desirable to “increase” frictional resistance, while in applications such as lubrication, it is desirable to decrease frictional resistance.

We apologize for using the simplified interpretation. The key goal of the work was to increase wear resistance of the polymer composites. The authors analyzed a change in the friction coefficients only in terms of interpreting the observed effects. The task of frictional resistance reducing was not considered in the manuscript.

Introduction, line 16: Expression “To achieve it” should be “To achieve this”.

Thanks for highlighting the mistake; it has been corrected in the revised edition of the manuscript.

Abstract, lines 17-19: Sentence should be corrected as “the effects…. were evaluated”

Thanks for highlighting the mistake; it has been corrected in the revised edition of the manuscript.

Line 43: Please check and use preposition “the” when applicable (it is missing from many parts of the text. An example in line 43: “This expands the applications …”

Thanks for highlighting the mistake; it has been corrected in the revised edition of the manuscript.

Line 258: The expression should be corrected as “after the end of the tribological tests”.

Thanks for highlighting the mistake; it has been corrected in the revised edition of the manuscript.

Line 325: The expression should be corrected as “The results of friction coefficient measurements”.

Thanks for highlighting the mistake; it has been corrected in the revised edition of the manuscript.

Reviewer 2 Report

The work is well-written and the results are interesting. I therefore recommend the publication with addressing the minor point below:

the quality of the plots 6, 7, 8, 9, 11 and 12 is rather low. Figure 17 is kinda convoluted too. Replace these with better quality plots.

Author Response

Dear Reviewer, greatest thanks for your efforts to improve the manuscript. please find enclosed some of our remarks. The corrections in the revised manuscript are marked with cyan.

the quality of the plots 6, 7, 8, 9, 11 and 12 is rather low. Figure 17 is kinda convoluted too. Replace these with better quality plots.

Unfortunately, the quality of the illustrations deteriorated sharply during the conversion of the manuscript to the 34 MB PDF file. In its original format, the figures fully meet the requirements of the journal.

Reviewer 3 Report

The paper si quite interesting and belongs to journal scope. 

Some comments that may helps to improve the quality of this.

Which application do you wnat to improve in particular because is difficult to undersatnd from text 

Was better to put the experiments firstly and the simulation

The finite elemement methods is very brieftly discussed and presented, it require more details in order to replicate the simulations

How the filler were disprsed in simulation ?

It is stated that 60 N is moderate loading condition, I am not sure if taht menas moderate as long as you don't suggets an application to indicate why is moderate

In which consideration was selected the percentage of carbon fibers?

Little bit confused why were used "Pin-on-disk" and "block-on-ring" please clarify it

Which was the temperature at the begining of test ? An increase of temperature with max 20 degree do not affect much the system (Fig 7)

From your Figure 14 I think the standard deviation is higher for UHMWPE + 10% LCF not for blue one.

Author Response

Dear Reviewer, greatest thanks for your efforts to improve the manuscript. Please find enclosed some of our remarks. The corrections in the revised manuscript are marked with cyan.

The paper is quite interesting and belongs to journal scope.

Some comments that may help to improve the quality of this.

- Which application do you want to improve in particular because is difficult to understand from text

The studies were devoted to the development of the wear-resistant polymer-matrix composites for operation in the dry friction conditions and in the wide temperature range (from –80 to +80 °C). Loading with inexpensive reinforcing fillers was designed to improve their mechanical properties and wear resistance at the same time. Promising areas of their practical use are, for example, the manufacture of lining plates for construction and marine equipment, chippers for transport infrastructure, etc.

This has been added in the revised edition of the manuscript.

- Was better to put the experiments firstly and the simulation

Your comment is very relevant but the authors ask to keep the current manuscript order for the following reason. In the presented form of the manuscript, the idea of the theoretical section is connected with the substantiation of the possibility of using reinforcing inclusions, as well as the explanation of the mechanism of their effect on wear resistance improving. Then, the analysis of their effect on wear resistance of the composites loaded with the fibers of various sizes was made using different load–speed parameters. This study can be considered as parametric, i.e. summarized the manuscript idea. If the theoretical section was in the final part of the manuscript, it should quantitatively describe all experimentally found results. Unfortunately, this cannot be fully done at present.

This has been added in the revised edition of the manuscript.

- The finite element methods is very briefly discussed and presented, it require more details in order to replicate the simulations.

Dear reviewer, your comment is very appropriate. Additional references and a brief description of the finite element method have been added in the revised edition of the manuscript. Moreover, the authors previously published an article describing the used approach in more detail (unfortunately, not in the English version of a journal). Thanks again for the remark and note that a separate article will be devoted to modeling aspects of the tribological contact processes using the finite element method. This is currently at the preparing stage.

- How the filler were dispersed in simulation ?

Inclusions were rectangular. Their location in the computational domain was set using a random number generator that determined the insertion point and the inclination angle. The ideal contact conditions were between the fibers and the matrix (added in the revised manuscript).

- It is stated that 60 N is moderate loading condition, I am not sure if that means moderate as long as you don't suggest an application to indicate why is moderate

You are absolutely right. The phrase “moderate loading conditions” meant the conditions of tribological loading that caused a low wear rate (comparable to that traditionally used in similar published studies), as well as not accompanied by a noticeable change in temperature of the counterpart. In this case, wear developed mainly by the fatigue mechanism, and there were no micro-grooves or deformations on the friction surface. Increases in load or speed under tribological loading caused rise in temperature and wear rate.

This has been added in the revised edition of the manuscript.

- In which consideration was selected the percentage of carbon fibers?

The authors tried to maintain a balance while improving both mechanical and tribological characteristics. To the greatest extent, it was determined by the distribution of the fibers in the polymer matrix, as well as the type of the formed permolecular structures.

- Little bit confused why were used "Pin-on-disk" and "block-on-ring" please clarify it

The existing tribometer made it possible to measure the friction coefficients. However, high specific pressures were in the tribological contact. It was shown that this method of UHMWPE wear resistance determining was not always sensitive enough to changes in its structure and mechanical properties. The friction machine that implements the “block-on-ring” test scheme did not enable to measure the friction coefficients. However, the load–speed parameters of the tribological tests could vary widely. In addition, it was possible to analyze in detail the changes on the friction surfaces, as well as the subsurface layer structure due to larger samples and lower specific pressure in the tribological contact.

This has been added to the revised edition of the manuscript.

- Which was the temperature at the beginning of the test ? An increase of temperature with max 20 degree do not affect much the system (Fig 7)

You are absolutely right. We started at room temperature. The temperature control method used by the authors was rather approximate, as was noted in the manuscript. However, the following temperature analysis principle was used. Almost no temperature changes occurred with the moderate loading conditions, and the authors did not even discuss its fluctuations within 2 ... 3 degrees, referred this to a possible dispersion or errors. Therefore, the authors considered tens of degrees as a significant change in temperature (it was taken into account when interpreting the results of changes in wear resistance). In this case, there were signs of strain intensification on the surface of the wear tracks due to visually observed frictional heating. We agree with the remark and a laboratory device is being produced for more accurate and reliable temperature measurement in tribological contacts.

This has been added to the revised edition of the manuscript.

- From your Figure 14 I think the standard deviation is higher for UHMWPE + 10% LCF not for blue one.

You are absolutely right. Standard deviation was maximum for curve 3 in Figure 14b, although it should have been for curve 4 according to Figure 14a.

It has been corrected in the revised edition of the manuscript

Round 2

Reviewer 3 Report

The authors responded to all reviewers comments, therefore I suggest acceptance of this manuscript

Author Response

Respectful Reviewer,

many thanks for your support of our study.

We have made a number of improvements in language sense. The modified manuscript has been resubmitted to the editorial board.

on behalf of the authors,

Sergey Panin